# First-Line Prescriptions and Effectiveness of *Helicobacter pylori* Eradication Treatment in Ireland over a 10-Year Period: Data from the European Registry on *Helicobacter pylori* Management (Hp-EuReg)

**DOI:** 10.3390/antibiotics14070680

**Published:** 2025-07-05

**Authors:** Sinéad M. Smith, Olga P. Nyssen, Rebecca FitzGerald, Thomas J. Butler, Deirdre McNamara, Asghar Qasim, Conor Costigan, Anna Cano-Catalá, Pablo Parra, Leticia Moreira, Francis Megraud, Colm O’Morain, Javier P. Gisbert

**Affiliations:** 1Trinity Academic Gastroenterology Group (TAGG) Research Centre, School of Medicine, Trinity College Dublin, D02 PN40 Dublin, Ireland; fitzger4@tcd.ie (R.F.); thbutler@tcd.ie (T.J.B.); mcnamad@tcd.ie (D.M.); colmomorain@gmail.com (C.O.); 2Hospital Universitario de La Princesa, Instituto de Investigación Sanitaria Princesa (IIS-Princesa), Universidad Autónoma de Madrid (UAM) and Centro de Investigación Biomédica en Red de Enfermedades Hepáticas y Digestivas (CIBERehd), 28006 Madrid, Spain; opn.aegredcap@aegastro.es (O.P.N.); pablo.parra.hlp@gmail.com (P.P.); javierpgisbert@gmail.com (J.P.G.); 3Department of Gastroenterology, Tallaght University Hospital, D24 NR0A Dublin, Ireland; 4Department of Gastroenterology, Beacon Hospital, D18 AK68 Dublin, Ireland; asgharqasim@gmail.com; 5Gastrointestinal Oncology, Endoscopy and Surgery Research Group, Althaia Xarxa Assistencial Universitària de Manresa, 08243 Manresa, Spain; acano@aegastro.es; 6Hospital Clínic de Barcelona, Centro de Investigación Biomédica en Red en Enfermedades Hepáticas y Digestivas (CIBERehd), IDIBAPS (Institut d’Investigacions Biomèdiques August Pi i Sunyer), University of Barcelona, 08007 Barcelona, Spain; 7Institut National de la Santé Et de la Recherche Médicale U1053 Bordeaux Institute of Oncology, Université de Bordeaux, 33076 Bordeaux, France; francis.megraud@u-bordeaux.fr

**Keywords:** amoxicillin, antibiotic resistance, bismuth, clarithromycin, first-line treatment, *Helicobacter pylori*, Ireland

## Abstract

**Background:** Local audits of *Helicobacter pylori* (*H. pylori*) prescriptions and outcomes are necessary to assess guideline awareness among clinicians and treatment effectiveness. **Aims:** The aims were to investigate first-line prescriptions and effectiveness over a 10-year period in Ireland and evaluate the influence of the 2017 Irish consensus guidelines on these trends. **Methods:** Data were collected at e-CRF AEG-REDCap from the European Registry on *H. pylori* management (Hp-EuReg) and quality reviewed from 2013 to 2022. All treatment-naïve cases were assessed for effectiveness by modified intention-to-treat (mITT) analysis. Multivariate analysis was also performed. **Results:** Data from 1000 patients (mean age 50 ± 15 years; 54% female) were analyzed. Clarithromycin (C) and amoxicillin (A) triple therapy represented 88% of treatments, followed by sequential C, A, and metronidazole (M) therapy (4.3%) and triple C + M (2.7%). Bismuth quadruple therapy was prescribed in 1.7% of cases. Treatment durations of 14, 10, and 7 days accounted for 87%, 4.5%, and 8.5% of prescriptions, respectively. High-, standard-, and low-dose proton pump inhibitors (PPIs; 80 mg, 40 mg, and 20 mg omeprazole equivalent b.i.d.) were used in 86%, 0.9%, and 13% of cases, respectively. The overall eradication rate was 80%, while it was 81% for triple C + A. Good compliance and high-dose PPI were associated with higher overall mITT eradication rates (OR 4.5 and OR 1.9, respectively) and triple C + A eradication rates (OR 4.2 and OR 1.9, respectively). Overall eradication rates increased from 74% pre-2017 to 82% (*p* < 0.05) by the end of 2022. Similarly, the triple C + A eradication rates increased from 76% to 83% (*p* < 0.05). **Conclusions:** While first-line treatment effectiveness improved in clinical practice over time, cure rates remain below 90%. Alternative first-line strategies are required in Ireland.

## 1. Introduction

The Gram-negative bacterium *Helicobacter pylori* (*H. pylori*) infects the stomach of almost half of the world’s population [1,2], leading to chronic gastritis and increasing the risk of developing peptic ulcers, gastric adenocarcinoma, and gastric mucosa-associated lymphoid tissue lymphoma [3,4,5,6,7]. Unexplained iron-deficiency anemia, vitamin B12 deficiency, and some cases of idiopathic thrombocytopenic purpura are also linked with *H. pylori* infection [3,8,9,10,11]. *H. pylori* is diagnosed non-invasively by the urea breath test, stool antigen test, or serology, or invasively by endoscopic tests, such as the rapid urease test, histology and bacterial culture, or PCR [3,8,12]. Treatment most commonly involves a proton pump inhibitor (PPI) and 2–3 antimicrobials [8,13,14,15,16,17,18,19,20,21]. However, treatment failure has become more frequent over time, mainly due to antimicrobial-resistant *H. pylori*, especially with regard to clarithromycin [22,23,24,25,26,27,28,29]. As the prevalence of *H. pylori* resistance varies geographically [23,30,31,32,33,34,35,36], local audits of *H. pylori* prescriptions and eradication rates are necessary to assess treatment efficacy in each population. In addition, audits are important to evaluate awareness and compliance with the most up-to-date clinical guidelines among prescribers.

The European registry on *H. pylori* management (Hp-EuReg) was established in 2013 to gather data on the diagnosis and treatment of *H. pylori* and perform time trend evaluations to improve the management of *H. pylori* in adult patients [37]. The Irish *H. pylori* working group was established in 2016 and published the first consensus recommendations on the management of *H. pylori* infection specific to the Irish healthcare setting in 2017 [38]. Clarithromycin (C) and amoxicillin (A) triple therapy or bismuth quadruple therapy (BQT; bismuth salt, tetracycline, metronidazole, PPI) were the recommended first-line treatment options. Other key recommendations included treatment durations of 14 days and the use of high-dose proton pump inhibitors (PPIs; 80 mg omeprazole equivalent bis in die (b.i.d.)) [38]. Thus, the aim of this study was to analyze the Irish data from the Hp-EuReg to (i) investigate first-line *H. pylori* prescription patterns and cure rates over a 10-year period and (ii) evaluate the impact of the first Irish consensus guidelines on these trends.

## 2. Results

### 2.1. Patient Characteristics

Of the cases included in the registry from Irish centers between 2013 and 2022, 1000 patients (mean age 50 ± 15 years; 54% (*n* = 536) female) were prescribed first-line therapy. In total, 78% (*n* = 784) were from Tallaght University Hospital, Dublin and 22% (*n* = 216) were from the Beacon Hospital, Dublin. The most common indication was dyspepsia (92%; *n* = 920), followed by duodenal ulcer and gastric ulcer (2%; *n* = 20 and 1%; *n* = 10, respectively) (Table 1). In total, 1.4% (*n* = 14) of patients reported penicillin allergy. Most patients (90%; *n* = 903) were not taking any concurrent medications. The rapid urease test was the most frequently performed for *H. pylori* diagnosis (47%; *n* = 465), while the urea breath test was carried out in 36% (*n* = 359) of cases (Table 1). Culture and antimicrobial susceptibility testing were performed in 3.1% cases (*n* = 31), with bacterial resistance rates of 23% (*n* = 7), 42% (*n* = 13) and 3% (*n* = 1) for clarithromycin, metronidazole, and levofloxacin, respectively (Table 1).

### 2.2. Prescription Patterns

C and A triple therapy was the most common first-line treatment overall (88%; *n* = 880) (Table 2). The second most prescribed treatment was for sequential therapy (PPI, A for 5d, followed by PPI, C, metronidazole (M) for 5d) at 4.3% (*n* = 43). C and M triple therapy was prescribed to 2.7% (*n* = 27) of patients, including those with penicillin allergy (*n* = 14; Table 1). Prescriptions for bismuth quadruple therapy (BQT) were infrequent at 1.7% (*n* = 17). There was heterogeneity among the BQT prescribed, with 14 cases of bismuth with M and tetracycline (T), 2 cases of bismuth with M and A, and 1 case of bismuth with C and T (Table 2). Figure 1 displays trends in prescription patterns over time. In all years except for 2014, triple C + A represented >85% of first-line prescriptions (Figure 1A). Reported compliance was high over the 10-year period at 99% (*n* = 986) (Table 2).

Of the reported treatment durations, 87% (*n* = 868/998) were for 14 days (Table 3). Of the reported PPI dose prescribed, 86% (*n* = 860/999) were high-dose (Table 3). Between 2013 and 2022, there was a trend towards increased treatment durations and PPI dose over time (Figure 1B,C).

Prescriptions for 14-day therapy significantly increased from 39% (*n* = 82/208) in the period of 2013–2016 to 99% (*n* = 786/790; *p* < 0.0001) in the period of 2017–2022 (Table 3). Similarly, the use of high-dose PPI significantly increased from 37% (*n* = 77/207) pre-2017 to 99% (*n* = 783/792; *p* < 0.0001) in the later years of the study (Table 3).

### 2.3. Effectiveness of First-Line Treatments

The overall eradication rate was 80% (95% CI 78–83%) and 81% (95% CI 79–84%) for triple C + A therapy (Table 4). By univariate analysis, 14-day prescriptions and higher-dose PPI were significantly associated with improved cure rates (Table 4; *p* < 0.005 and *p* < 0.0005, respectively). There was a trend towards increased effectiveness between 2013 and 2022 (Figure 2). In keeping with this, overall eradication rates significantly increased from 75% (95% CI 68–80%) pre-2017 to 82% (95% CI 79–84%; *p* < 0.05) (Table 5), while triple C + A therapy eradication rates significantly increased from 76% (95% CI 68–82%) pre-2017 to 83% (95% CI 80–85%; *p* < 0.05) afterwards.

Finally, multivariate regression analysis was performed to examine the impact of multiple variables on mITT cure rates from 2013 to 2022. Good compliance and the use of high-dose PPI were significantly associated with better overall mITT rates (OR 4.5, 95% CI 1.4–14.2 and OR 1.9, 95% CI 1.2–2.8, respectively) (Table 6). Good compliance and increased acid suppression were also significantly associated with the success of triple C + A therapy (OR 4.2, 95% CI 1.2–14.2 and OR 1.9, 95% CI 1.1–3.2, respectively) (Table 6). Age, sex, indication, and treatment duration were not significantly associated with an increase in treatment effectiveness by multivariate analysis.

## 3. Discussion

Previous studies on first-line prescriptions in countries included in the Hp-EuReg have shown that the management of *H. pylori* among prescribing gastroenterologists across Europe is heterogeneous [39]. Here we show that in Ireland, the most common first-line therapy between 2013 and 2022 was triple C + A (88%; *n* = 880/1000). The second most prescribed treatment was for sequential therapy at 4.3% (*n* = 43/1000). In 2014, these patients received sequential therapy as part of a prospective randomized controlled study that took place at Tallaght University Hospital that year comparing sequential therapy with triple C + A [40]. There was no significant difference in eradication rates between sequential therapy and triple C + A in that study [40], providing a possible explanation for the lack of sequential therapy prescribed in subsequent years of the current analysis. In 2017, Irish consensus guidelines for *H. pylori* were published, recommending treatment durations of 14 days and the use of high-dose PPIs [38]. Interestingly, when the data from 2013 to 2016 were compared to those from 2017 to 2022, prescriptions for 14-day therapy significantly increased from 39% to 99% (*p* < 0.0001), and the use of high-dose PPI significantly increased from 37% to 99% (*p* < 0.0001). While direct causality cannot be definitively established, these findings suggest both awareness and compliance with the guidelines among the recruiting gastroenterologists in our study.

Over the 10-year period analyzed, the overall eradication rate was 80% (95% CI 78–83%) and 81% (95% CI 79–84%) for triple C + A therapy, which was prescribed in almost 90% of cases. Univariate analysis showed that a longer treatment duration and higher-dose PPI were significantly associated with improved overall cure rates. By multivariate analysis, good patient compliance and high-dose PPI were associated with higher overall cure rates (OR 4.5, 95% CI 1.4–14.2 and OR 1.9, 95% CI: 1.2–2.8, respectively) and triple C + A cure rates (OR 4.2, 95% CI 1.2–14.2 and OR 1.9, 95% CI: 1.1–3.2, respectively). These findings correspond with what has been observed in studies from other countries [39,41,42,43,44,45,46,47,48,49].

In line with the increase in prescriptions for 14 days and the use of high-dose PPI in the period 2017–2022 compared to 2013–2016, overall eradication rates significantly increased from 75% (95% CI 68–80%) to 82% (95% CI 79–84%), while triple C + A eradication rates significantly increased from 76% (95% CI 68–82%) to 83% (95% CI 80–85%). Despite the enhanced eradication rates observed, they fall short of the 90% target cure rate for an optimized *H. pylori* therapy [8,39,50]. Given that reported compliance was high in our study, the most likely explanation for failed eradication therapy is the presence of *H. pylori* resistant to one or more of the antimicrobials used [51]. Primary resistance was >15% for both clarithromycin and metronidazole in the study cohort. While these figures are limited by the very small number of cases where culture and antimicrobial susceptibility testing were performed in the study population (3.1%), these resistance rates are in line with the Irish data published in the pan-European resistance study [30] and other European studies where resistance rates of >15% for clarithromycin and metronidazole among treatment-naïve patients were described [31]. The number of cases where antimicrobial susceptibility testing was performed during the study was disappointing, given that resistance surveillance is recommended [8,30].

According to European guidelines, first-line triple C + A therapy should not be used in areas with unknown clarithromycin resistance or resistance >15% [8]. Other treatment options include BQT (a bismuth salt, tetracycline, metronidazole, and PPI), non-bismuth quadruple (concomitant) therapy (clarithromycin, metronidazole, amoxicillin, and PPI), and high-dose PPI plus amoxicillin dual therapy. Concomitant therapy is unlikely to be suitable in our country due to high rates of primary dual clarithromycin and metronidazole resistance. Further, the eradication rate for high-dose PPI plus amoxicillin dual therapy in our country is poor [52]. In the current study, the only treatment to achieve an eradication rate above 90% was BQT (94%). However, this finding is limited by the extremely low number of patients that were prescribed this therapy (1.7%). This was likely due to difficulty accessing bismuth salts in Ireland during the timeframe of the study. In Ireland, De-Noltab (bismuth subcitrate potassium 120 mg) is an Exempt Medicinal Product (EMP), and pharmacists are required to obtain this medication from a special wholesaler. The combination 3-in-1 capsule Pylera (140 mg bismuth subcitrate potassium, 125 mg metronidazole, and 125 mg tetracycline hydrochloride) is licensed but not marketed in Ireland, making it challenging for pharmacies to source [38]. Despite this, a strong rationale for the use of BQT in European populations has been provided by other studies from the Hp-EuReg, which consistently report BQT eradication rates of >90% [37,39].

Another potential strategy to enhance eradication rates involves the use of the potassium competitive acid blocker vonoprazan instead of the PPI in anti-*H. pylori* therapies, with meta-analyses showing higher eradication rates for vonoprazan-based triple or quadruple therapies compared to PPI-based therapies [53,54,55,56]. Notwithstanding these encouraging findings, vonoprazan is not available in Ireland at the present time.

The strengths of this study are that the analysis represents the largest audit of *H. pylori* treatment undertaken in Ireland to date and includes 1000 patients from both the public (Tallaght University Hospital) and private healthcare setting (Beacon Hospital). Patients from both urban and rural communities are referred to these hospital sites and were recruited over a decade. However, a limitation of the study is that both hospitals are located in Co. Dublin, with patient referrals mainly from the east of the country. To overcome this limitation, the authors acknowledge the importance of including additional recruiting hospitals across the island of Ireland going forward in order to more accurately assess prescription patterns and cure rates in a population more representative of the country in its entirety. Prescribing practices in the primary care setting would also be of great interest to assess trends and guideline awareness among primary care physicians. Further, the impact of antimicrobial resistance on first-line treatment outcomes in the study population is limited by the low number of isolates that underwent antimicrobial susceptibility testing. Efforts should be made to increase antimicrobial susceptibly testing, either by culture or molecular methods, for ongoing resistance surveillance and a more thorough evaluation of the impact of resistance on treatments prescribed.

In summary, while it is encouraging that eradication rates for the most prescribed first-line therapy (triple C + A) have improved over time, they are still sub-optimal. The cure rates reported herein, the recent Irish and European data on primary clarithromycin resistance [30,31], together with the superior cure rates reported for BQT throughout Europe [37,39], support the use of BQT in the Irish healthcare setting. Indeed, the Irish consensus guidelines have recently been updated to recommend BQT in our population unless pre-treatment clarithromycin susceptibility testing is available and susceptibility is confirmed [57]. In relation to bismuth products, De-Noltab has become easier to obtain by pharmacists, and the use of BQT as first-line anti-*H. pylori* treatment is actively being promoted to prescribers through presentations at relevant national society conferences for gastroenterologists and primary care doctors, as well as through continuous medical education programs. Ongoing participation in the Hp-EuReg through current and additional Irish centers will be imperative to monitor the effects of the updated recommendations and the efficacy of BQT in our population going forward.

## 4. Materials and Methods

The Hp-EuReg is an international multicenter prospective non-interventional registry, collecting information on the management of *H. pylori* infection from 38 countries since 2013. The study protocol [58] conforms to the ethical guidelines of the 1975 Declaration of Helsinki as reflected in a prior approval by the institution’s human research committee. The study was classified by the Spanish Agency for Medicines and Health Products, prospectively registered in ClinicalTrials.gov (NCT02328131), and was approved by the Ethics Committee of La Princesa University Hospital, Madrid, Spain.

*H. pylori*-positive adult patients attending gastroenterology out-patient clinics for *H. pylori* testing at Tallaght University Hospital and the Beacon Hospital in Ireland and who were prescribed anti-*H. pylori* therapy were enrolled in the Hp-EuReg. The current study is a sub-analysis of treatment-naïve cases enrolled at the 2 Irish centers from 2013 to 2022. Each country in the Hp-EuReg has a national coordinator. Author SMS is the national coordinator for Ireland and was responsible for supervising data inclusion and drafting this study. The recruiting investigators were gastroenterologists and/or members of their teams. Patients were managed and registered following routine clinical practice. Irish-based investigators were authors RF, TJB, DMN, AQ, CC, and COM.

### 4.1. Data Management

Data were recorded in an electronic Case Report Form (e-CRF) using a web-based application RED-Cap (Research Electronic Data Capture), a platform managed and hosted by the non-profit Scientific and Medical Society Asociaciόn Española de Gastroenterología” (AEG; www.aegastro.es, accessed on 2 July 2025). Patient demographics, *H. pylori* treatment history, treatments prescribed, and outcomes (eradication rates, compliance) were recorded. After extracting the data and prior to the statistical analysis, the database was reviewed for quality, resolving inconsistencies. For the current study, treatment-naïve cases collected in Ireland between June 2013 and December 2022 were evaluated.

### 4.2. Variable Categorization and Definitions

Seven different treatment schemes were used in first-line therapy: triple C + A (a PPI together with clarithromycin and amoxicillin), sequential C + A + M (a PPI together with clarithromycin, amoxicillin and metronidazole given in a sequential way), triple C + M (a PPI together with clarithromycin and metronidazole), triple L + A (a PPI together with levofloxacin and amoxicillin), BQT (a PPI together with a bismuth salt and 2 antimicrobials), triple M + A (a PPI together with metronidazole and amoxicillin), and triple C + L (a PPI together with clarithromycin and levofloxacin). Antibiotic doses were as follows: clarithromycin 500 mg b.i.d., amoxicillin 1g b.i.d., metronidazole 400 mg b.i.d. (or t.i.d. when used in BQT), and 250 mg levofloxacin, b.i.d. PPI data were standardized using the PPI acid inhibition potency [39] and classified as low-dose (20 mg omeprazole equivalents b.i.d.), standard-dose (40 mg omeprazole equivalents b.i.d.), and high-dose (80 mg omeprazole equivalents b.i.d). Treatment durations were analyzed according to prescriptions for 7, 10, or 14 days.

The main outcome (i.e., eradication success) was confirmed *H. pylori* eradication at least 4 weeks after treatment. All treatment-naïve cases were assessed for effectiveness by modified intention-to-treat (mITT) analysis. The mITT includes all cases that completed a follow-up valid confirmatory test at least 4 weeks after their *H. pylori* treatment, regardless of compliance and excluding those lost to follow-up. This method of analysis was selected with the aim of most accurately reflecting what happens in real-world clinical practice and is the form of analysis used to calculate cure rates in other studies published from the Hp-EuReg [37,39]. All patients who were empirically treated (i.e., not susceptibility-guided) were included in the effectiveness analysis. Patient compliance with treatment was defined as more than 90% drug intake and was based on self-reported patient compliance.

### 4.3. Statistical Analysis

Continuous variables are presented as arithmetic means and standard deviations. Qualitative variables are presented as absolute and relative frequencies. Univariate analyses were conducted based on the overall first-line prescriptions, accounting for the PPI dosage and treatment duration. The two-tailed χ^2^ test was used to compare differences between groups. Multivariate analysis was performed using a logistic regression model to evaluate the relationship between mITT rates (as the dependent variable) and the independent variables age, sex, indication, compliance, PPI dose, and treatment duration. Odds ratios (ORs) and their 95% confidence intervals (CIs) are provided. For all statistical analyses undertaken, a *p* < 0.05 was considered significant.

## 5. Conclusions

Triple C + A therapy was the most frequently prescribed first-line treatment for *H. pylori* between 2013 and 2022 at Irish centers participating in the Hp-EuReg. Although there was an increase in the first-line eradication rate over time, treatment was unsuccessful in more than 10% of cases, providing a rationale for a different approach to the management of *H. pylori* in treatment-naïve patients in Ireland.

## Figures and Tables

**Figure 1 antibiotics-14-00680-f001:**
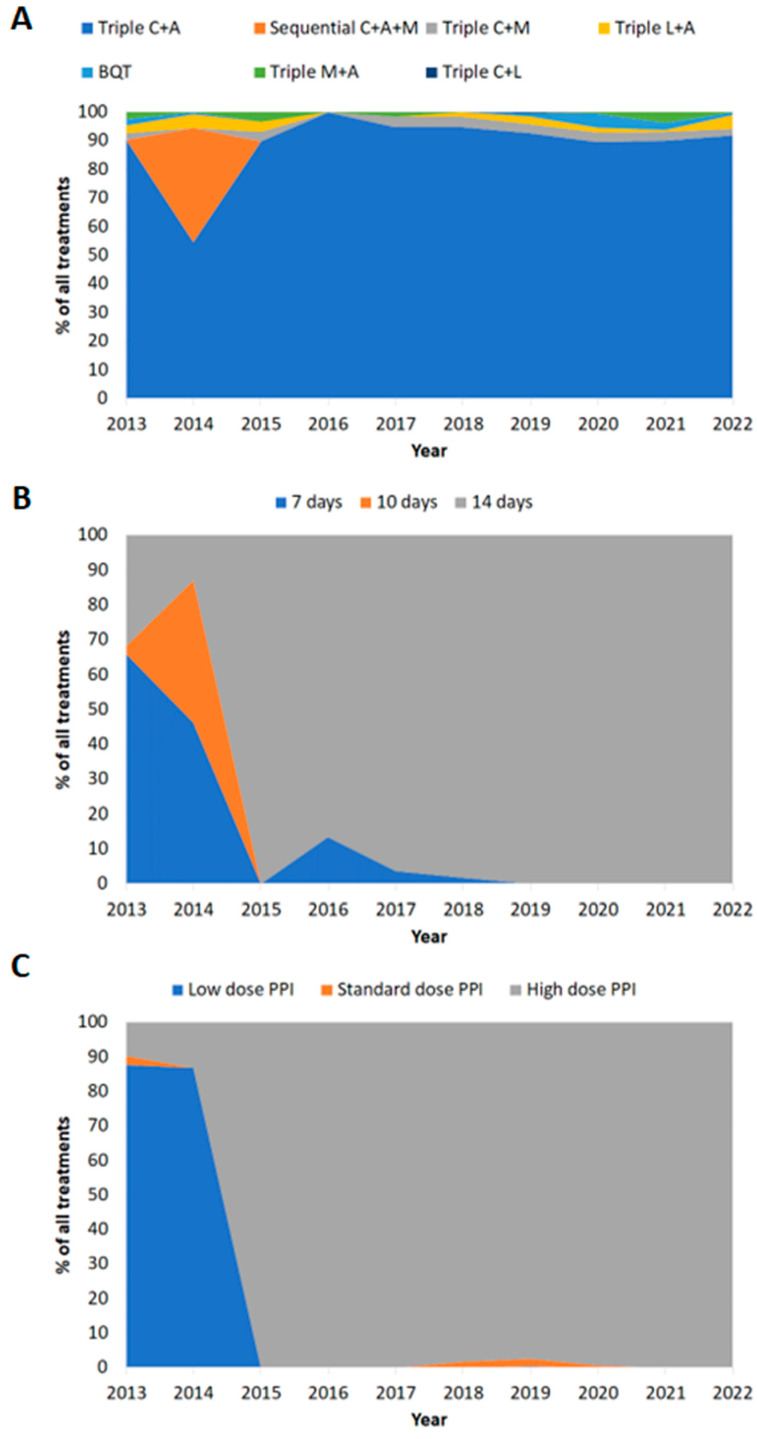
Prescription patterns in Ireland over time (2013–2022). (**A**) Treatment regimen prescribed. (**B**) Treatment duration. (**C**) Dose of PPI prescribed. C: clarithromycin; A: amoxicillin; M: metronidazole; L: levofloxacin; BQT: bismuth quadruple therapy; low-dose PPI: 20 mg omeprazole equivalent twice daily; standard-dose PPI: 40 mg omeprazole equivalent twice daily; high-dose PPI: 80 mg omeprazole equivalent twice daily.

**Figure 2 antibiotics-14-00680-f002:**
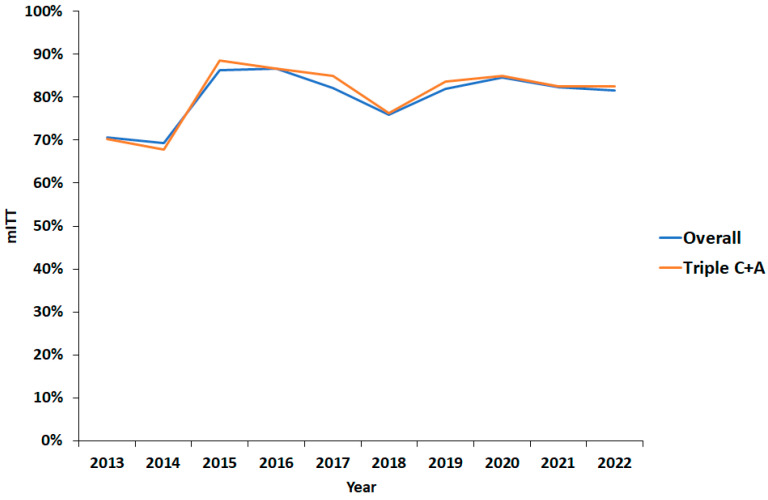
Effectiveness by modified intention-to-treat eradication rate over time. mITT: modified intention-to-treat; C: clarithromycin; A: amoxicillin.

**Table 1 antibiotics-14-00680-t001:** Baseline characteristics of patients prescribed empirical first-line *H. pylori* eradication therapies.

Variable	*n*	%
**Female**	536	53.6
**Indication**		
Dyspepsia	920	92.0
Duodenal ulcer	20	2.0
Gastric ulcer	10	1.0
Other	50	5.0
**Penicillin allergy**	14	1.4
**Concurrent medication**		
None	903	90.3
PPI	77	7.7
Acetylsalicylic acid	63	6.3
Statins	62	6.2
NSAIDs	54	5.4
Unknown	5	0.5
**Diagnosis**		
Rapid urease test	465	46.5
Histology	370	37.0
Urea breath test	359	35.9
Other	5	0.5
Culture	31	3.1
No resistance	13	41.9
ClarR	7	22.6
MetR	13	41.9
LevoR	1	3.2

PPI: proton pump inhibitor; NSAIDs: non-steroidal anti-inflammatory drugs; ClarR: clarithromycin-resistant; MetR: metronidazole-resistant; LevoR: levofloxacin-resistant.

**Table 2 antibiotics-14-00680-t002:** First-line treatments prescribed from 2013 to 2022.

Variable	Overall	
	*n*	%
**Treatment**		
Triple C + A	880	88.0
Sequential C + A + M	43	4.3
Triple C + M	27	2.7
Triple L + A	22	2.2
BQT ^1^	17	1.7
Triple M + A	10	1
Triple C + L	1	0.1
**Compliance**	986	98.6

C: clarithromycin; A: amoxicillin; M: metronidazole; L: levofloxacin; BQT: bismuth quadruple therapy; ^1^ 14 cases of bismuth (B) + tetracycline (T) + M, 2 cases of B + M + A, 1 case of B + C + T.

**Table 3 antibiotics-14-00680-t003:** First-line treatment durations and proton pump inhibitor dose prescribed.

Variable	Overall		2013–2016		2017–2022		*p*-Value ^3^
	*n*	%	*n*	%	*n*	%	
**Duration ^1^**							
7 days	85	8.5	81	38.9	4	0.5	<0.0001
10 days	45	4.5	45	21.6	0	0.0	<0.0001
14 days	868	86.8	82	39.4	786	99.0	<0.0001
**PPI dose ^2^**							
Low	130	13	129	62.3	1	0.1	<0.0001
Standard	9	0.9	1	0.5	8	1.0	0.475
High	860	86.1	77	37.2	783	98.9	<0.0001

^1^ Treatment duration was known in 998 cases; ^2^ PPI dose was known in 999 cases; ^3^ results from 2013–2016 versus 2017–2022 were compared using the two-tailed χ^2^ test. Low-dose PPI: 20 mg omeprazole equivalent twice daily; standard-dose PPI: 40 mg omeprazole equivalent twice daily; high-dose PPI: 80 mg omeprazole equivalent twice daily.

**Table 4 antibiotics-14-00680-t004:** Eradication success by treatment, duration, and proton pump inhibitor dose.

Variable	mITT Effectiveness	
	*n*	% (95% CI)
**Treatment**		
Triple C + A	716	81.4 (78.7–83.8)
Sequential C + A + M	30	69.8 (54.8–81.5)
Triple C + M	15	55.6 (37.3–72.4)
Triple L + A	16	72.7 (51.6–87.1)
BQT ^1^	16	94.1 (71.1–100)
Triple M + A	8	80.0 (47.9–95.4)
Triple C + L	0	0.0 (0.0–8.3)
**Overall**	**801**	**80.1 (77.5–82.5)**
**Duration** ^2,4^		
7 days	57	67.1 (56.5–76.2)
10 days	32	71.1 (56.5–82.4)
14 days	710	81.8 (79.1–84.2)
**PPI dose** ^3,4^		
Low	88	67.7 (59.2–75.1)
Standard	6	66.7 (35.1–88.3)
High	706	82.1 (79.4–84.5)

mITT: modified intention-to-treat; 95% CI: 95% confidence interval; C: clarithromycin; A: amoxicillin; M: metronidazole; L: levofloxacin; BQT: bismuth quadruple therapy; ^1^ 14 cases of bismuth (B) + tetracycline (T) + M, 2 cases of B + M + A, 1 case of B + C + T; ^2^ treatment duration was known in 998 cases; ^3^ PPI dose was known in 999 cases; ^4^ results comparing mITT according to treatment duration or PPI dose were analyzed using the two-tailed χ^2^ test; *p* = 0.002 and *p* = 0.0003, respectively.

**Table 5 antibiotics-14-00680-t005:** Modified intention-to-treat effectiveness before and after Irish *Helicobacter pylori* working group consensus guidelines.

Variable	2013–2016		2017–2022		*p*-Value
	*n*	% (95% CI)	*n*	% (95% CI)	
All treatments	155	74.5 (68.2–80.0)	646	81.6 (78.7–84.1)	0.023
Triple C + A	115	75.7 (68.2–81.8)	601	82.6 (79.6–85.1)	0.047

95% CI: 95% confidence interval; C: clarithromycin; A: amoxicillin.

**Table 6 antibiotics-14-00680-t006:** Multivariate analysis of first-line modified intention-to-treat effectiveness.

Independent Variables	All Treatments		Triple C + A	
	OR (95% CI)	*p*-Value	OR (95% CI)	*p*-Value
Compliance ^1^	4.5 (1.4–14.2)	0.01	4.2 (1.2–14.2)	0.02
PPI dose ^2^	1.9 (1.2–2.8)	0.004	1.9 (1.1–3.2)	0.03

OR: odds ratio; 95% CI: 95% confidence interval; C: clarithromycin; A: amoxicillin; PPI: proton pump inhibitor; ^1^ ≥90% drug intake versus <90% drug intake; ^2^ high-dose PPI versus low and standard dose.

## Data Availability

Data associated with the study are included in this manuscript.

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
