# Peer review of "First-Line Prescriptions and Effectiveness of Helicobacter pylori Eradication Treatment in Ireland over a 10-Year Period: Data from the European Registry on Helicobacter pylori Management (Hp-EuReg)"

_antibiotics, 2025, doi:10.3390/antibiotics14070680_

Round 1
Reviewer 1 Report
Comments and Suggestions for Authors
Summary
This study addressed the effectiveness of H. pylori eradication treatments in Ireland over a 10 year period.
The authors specifically examined whether the 2017 Irish Consensus Guidelines led to changes in clinical practice.
Major comments
1. The authors conclusion that the 2017 Irish guidelines lead to increased prescription of 14 day regimens and high dose PPI is debatable. Figure 1 indicates that these changes began in 2015.
2. The authors should mention whether the patients enrolled in the registry can represent the Irish population.
Enrollment seems to have occurred in two hospitals.
3. The authors’ reasoning for assessing treatment based on mITT should be presented.
4. What was the reason for including 1000 patients?
5. The authors should present further data on patient enrollment.
Minor comments
1. As the authors mention the importance of high dose PPIs for eradication, they should present the types of PPIs used and their doses (Were only omeprazole used?)
2. Likewise, the authors should present the doses of clarithromycin, amoxicillin, and metronidazole for different regimens.
3. Did every patient receive the same dose of these antibiotics?
Author Response
Reviewer 1 Major Comment 1
The authors conclusion that the 2017 Irish guidelines lead to increased prescription of 14-day regimens and high-dose PPI is debatable. Figure 1 indicates that these changes began in 2015.
Response:
We thank the reviewer for this important point. The description of the analysis has been edited throughout to indicate that the data were compared between the time periods 2013-2016 and 2017-2022. The first paragraph of the Discussion has been edited to acknowledge that direct causality cannot be definitively established. However, the changes over time suggest awareness of the guidelines.
Reviewer 1 Major Comment 2
The authors should mention whether the patients enrolled in the registry can represent the Irish population. Enrollment seems to have occurred in two hospitals.
Response: The authors consider that the Irish population is well represented in the analysis. Patients were enrolled from Tallaght University Hospital, which is a public hospital, and the Beacon Hospital, which is a private hospital. Patients from both urban and rural communities are referred to these hospital sites. These details have now been included in the Discussion
Section in the new paragraph outlining the strengths and weaknesses of the study. In response to the reviewer’s comment, it is also noted in this paragraph that the study would be strengthened by recruitment from additional hospital sites around the country and from the primary care setting and this is planned as part of our on-going research in this area.
Reviewer 1 Major Comment 3
The authors’ reasoning for assessing treatment based on mITT should be presented.
Response: The treatment-naïve cases were assessed for effectiveness by modified intention- to-treat (mITT) analysis, which includes all cases that completed a follow-up valid confirmatory test at least 4 weeks after their H. pylori treatment, regardless of compliance and excluding those lost to follow up. This method of analysis was chosen to most closely represent what
happens in clinical practice and is in line with the analysis of other studies published from the Hp-EuReg [1-4]. This has been included in the Variable Categorisation and Definitions section of the Methods in the revised manuscript.
Reviewer 1 Major Comment 4
What was the reason for including 1000 patients?
Response: The analysis aimed to include all treatment-naïve patients recruited to the study between 2013-2022, rather than aiming for a total number of 1000. Of the cases recruited to the registry from Irish centres during this timeframe, 1000 were prescribed first-line treatment. This is now mentioned in the Patient Characteristics section of the Results.
Reviewer 1 Major Comment 5.
The authors should present further data on patient enrollment.
Response: H. pylori-positive adult patients attending gastroenterology out-patient clinics for H. pylori testing at Tallaght University Hospital and the Beacon Hospital in Ireland and who were prescribed anti-H. pylori therapy were enrolled in the Hp-EuReg. Patients were managed and registered following routine clinical practice. The current study is a sub-analysis of
treatment-naïve cases enrolled at the 2 Irish centres from 2013-2022. This information is now included in the Methods section.
Reviewer 1 Minor Comment 1
As the authors mention the importance of high dose PPIs for eradication, they should present the types of PPIs used and their doses (Were only omeprazole used?)
Response: There were variations in the type and dose of PPI used. For this reason, all PPI doses mentioned in the manuscript are described in terms of omeprazole equivalents and were standardised using the PPI acid inhibition potency [5]. They were classified as low dose (20 mg omeprazole equivalents bid), standard dose (40 mg omeprazole equivalents bid) and high
dose (80 mg omeprazole equivalents bid). These standards are described in the Abstract, included in the Variable Categorisation and Definitions Section of the Methods, and are also provided as footnotes in Figure 1 and Table 3. This PPI dose standardisation has also been used in other studies published by the Hp-EuReg (e.g. [1-4]).
Reviewer 1 Minor Comment 2
Likewise, the authors should present the doses of clarithromycin, amoxicillin, and metronidazole for different regimens.Did every patient receive the same dose of these antibiotics?
Response: The doses of these antibiotics were consistent throughout the study and are now included in the Variable Categorisation and Definitions section of the Methods.
Reviewer 2 Report
Comments and Suggestions for Authors
The manuscript presents a real-world analysis of Helicobacter pylori eradication therapies in Ireland over a decade, based on data from the Hp-EuReg. The study highlights changes in prescription patterns following the implementation of Irish consensus guidelines and examines the effectiveness of first-line treatments.
Some points that need to address:
Percent match is 45% in the iThenticate report. The authors might copy and paste some content directly from other publications. Please fix this!!!
The result section in Abstract: add other therapies
1.4% (n=14) of patients reported penicillin allergy. How are they treated?
Consider discuss vonoprazan-based treatment in the discussion part. Consider cite this article: Liu, L., Shi, H., Shi, Y., Wang, A., Guo, N., Li, F., & Nahata, M. C. (2024). Vonoprazan‐based therapies versus PPI‐based therapies in patients with H. pylori infection: Systematic review and meta‐analyses of randomized controlled trials. Helicobacter, 29(3), e13094.
Please add the eradication rate and compliance rate for certain therapy with duration in the result section. Perform sensitivity analyses considering additional variables to enhance the robustness of the results. For example: what is it for PAC 7 days? PAC 10 days? PAC 14 days? What is the dose of amoxicillin and clarithromycin? Are they the same? The dose of amoxicillin also significantly impact the eradication rate.
The data is collected from two major hospitals in Dublin (Tallaght University Hospital and Beacon Hospital), which may not fully represent the entire Irish population. The authors should discuss potential biases due to limited geographic representation.
Consider discussing why Bismuth quadruple therapy (BQT), despite higher efficacy, is underused and how to overcome accessibility challenges.
"The reason for the increase in sequential therapy that year was due to a prospective random
ized-controlled study that took place at Tallaght University Hospital comparing sequential therapy with triple C+A [19]." This should be in discussion part.
Line 208-212. Please give me more details about the guideline, what are the recommendations?
Clarify Causality: Emphasize that while guideline implementation correlates with improved eradication rates, causality cannot be definitively established.
compliance was high over the 10-year period at 99%. Why? It sounds not real.
Add a strength and limitation section for this manuscript.
Author Response
The manuscript presents a real-world analysis of Helicobacter pylori eradication therapies in Ireland over a decade, based on data from the Hp-EuReg. The study highlights changes in prescription patterns following the implementation of Irish consensus guidelines and examines the effectiveness of first-line treatments. Some points that need to address:
Reviewer 2 Comment 1
Percent match is 45% in the iThenticate report. The authors might copy and paste some content directly from other publications. Please fix this!!!
Response: We have tried our very best to reduce the similarity score in the revised manuscript. As mentioned to the Editorial Office Staff, there is similarity between the manuscript and the PhD thesis of author Rebecca FitzGerald as some (but not all) of the data are included in her PhD thesis.
In addition, some of the data in the manuscript were presented as posters at United European Gastroenterology Week 2024 and the European Helicobacter and Microbiota Study Group Workshop 2024, with conference abstracts from these meetings published in the UEG Journal and Microbiota and Health and Disease, respectively. We confirm that the submitted manuscript has not been published in other journals in its current form.
Reviewer 2 Comment 2
The result section in Abstract: add other therapies
Response: We have not provided extensive detail in the abstract about the other treatments due to the Abstract word limit and the fact that the numbers prescribed treatments apart from triple C+A were small. Nevertheless, some additional information on other therapies has been included in the abstract.
Reviewer 2 Comment 3
1.4% (n=14) of patients reported penicillin allergy. How are they treated?
Response: These patients were treated with clarithromycin and metronidazole triple therapy. This is now mentioned in the Prescription Patterns section of the Results.
Reviewer 2 Comment 4
Consider discuss vonoprazan-based treatment in the discussion part. Consider cite this article: Liu, L., Shi, H., Shi, Y., Wang, A., Guo, N., Li, F., & Nahata, M. C. (2024). Vonoprazan- based therapies versus PPI-based therapies in patients with H. pylori infection: Systematic review and meta-analyses of randomized controlled trials. Helicobacter, 29(3), e13094.
Response: Vonoprazan-based treatment is now addressed in the Discussion together with the suggested citation. Additional relevant references have also been cited. Unfortunately, vonoprazan is not available in our country.
Reviewer 2 Comment 5
Please add the eradication rate and compliance rate for certain therapy with duration in the result section. Perform sensitivity analyses considering additional variables to enhance the robustness of the results. For example: what is it for PAC 7 days? PAC 10 days? PAC 14 days?
Response: Thank you for this suggestion. The majority of prescriptions were for clarithromycin (C) and amoxicillin (A) triple therapy (88%, n=880/1000). The numbers of patients in the remaining treatment groups were low (sequential therapy, n=43; triple C+M, n=27; triple L+A, n=22; BQT, n=17; triple M+A, n=10; triple C+L, n=1). In addition, the number of non-compliant
patients overall was low (n=14/1000). Therefore, the sub-analysis of the eradication rate and compliance rates for each individual treatment (except triple C+A) with duration did not provide any meaningful results at the small numbers tested. We have performed multivariate analysis to evaluate the relationship between overall eradication rates (as the dependent variable) and the independent variables age, sex, indication, compliance, PPI dose and treatment duration (described in Methods). Similarly, we have performed multivariate analysis using triple C+A eradication rate as the dependent variable and age, sex, indication, compliance, PPI dose and treatment duration as the independent variables. In both the overall and triple C+A analyses, we found significant associations between compliance and PPI dose and eradication rates (Shown in Table 6).
Indeed, because the majority of patients received triple C+A (88%), results from the overall analysis and triple C+A analysis are very similar. For example, the overall eradication rate was 80% and 81% for triple C+A therapy. By multivariate analysis, age, sex, indication and treatment duration were not significantly associated with treatment effectiveness either overall or for triple C+A (included at the end of the results section).
Reviewer 2 Comment 6.
What is the dose of amoxicillin and clarithromycin? Are they the same? The dose of amoxicillin also significantly impact the eradication rate.
Response: The doses of these antibiotics were consistent throughout the study (clarithromycin was 500 mg b.i.d. and amoxicillin was 1g b.i.d.). This is now included in the Variable Categorisation and Definitions section of the Methods.
Reviewer 2 Comment
The data is collected from two major hospitals in Dublin (Tallaght University Hospital and Beacon Hospital), which may not fully represent the entire Irish population. The authors should discuss potential biases due to limited geographic representation.
Response: Thank you for this important point. This is further discussed in the paragraph relating to strengths and weakness of the study in the Discussion.
Reviewer 2 Comment 8
Consider discussing why Bismuth quadruple therapy (BQT), despite higher efficacy, is underused and how to overcome accessibility challenges.
Response: Further information on bismuth accessibility has been included in the Discussion.
Reviewer 2 Comment 9
"The reason for the increase in sequential therapy that year was due to a prospective randomized-controlled study that took place at Tallaght University Hospital comparing sequential therapy with triple C+A [19]." This should be in discussion part.
Response: This has now been moved to the Discussion.
Reviewer 2 Comment 10
Line 208-212. Please give me more details about the guideline, what are the recommendations?
Response: The guidelines recommended treatment durations of 14 days and the use of high dose PPI (e.g. 40 mg esomeprazole b.i.d or equivalent). This is described in the Introduction and now included in the Discussion according to Reviewer 2’s recommendations.
Reviewer 2 Comment 11
Clarify Causality: Emphasize that while guideline implementation correlates with improved eradication rates, causality cannot be definitively established.
Response: Thank you, that causality cannot be definitely established is mentioned in the first paragraph of the Discussion.
Reviewer 2 Comment 12
Compliance was high over the 10-year period at 99%. Why? It sounds not real.
Response: Compliance data was based on self-reported patient compliance (now described in Methods section). The authors acknowledge that this may not be as accurate as compliance data from a clinical trial, where the actual number of tablets taken by each patient is monitored.
Reviewer 2 Comment 13
Add a strength and limitation section for this manuscript.
Response: A strength and limitation paragraph has been added to the Discussion.
Reviewer 2 Comment 12
Compliance was high over the 10-year period at 99%. Why? It sounds not real.
Response: Compliance data was based on self-reported patient compliance (now described
in Methods section). The authors acknowledge that this may not be as accurate as compliance
data from a clinical trial, where the actual number of tablets taken by each patient is
monitored.
Reviewer 2 Comment 13
Add a strength and limitation section for this manuscript.
Response: A strength and limitation paragraph has been added to the Discussion.
References
1. Olmedo, L.; Calvet, X.; Gene, E.; Bordin, D.S.; Voynovan, I.; Castro-Fernandez, M.; Pabon-Carrasco, M.; Keco-Huerga, A.; Perez-Aisa, A.; Lucendo, A.J.; et al. Evolution of the use, effectiveness and safety of bismuth-containing quadruple therapy for Helicobacter pylori infection between 2013 and 2021: results from the European registry on H. pylori management (Hp-EuReg). Gut 2024, doi:10.1136/gutjnl-2024-332804.
2. Rokkas, T.; Georgopoulos, S.; Michopoulos, S.; Ntouli, V.; Liatsos, C.; Puig, I.; Nyssen, O.P.; Megraud, F.; O'Morain, C.; Gisbert, J.P.; et al. Assessment of first-line eradication treatment in Greece: data from the European Registry on Helicobacter pylori management (Hp-EuReg). Ann Gastroenterol 2022, 35, 42-47, doi:10.20524/aog.2021.0670.
3. Jonaitis, P.; Kupcinskas, J.; Nyssen, O.P.; Puig, I.; Gisbert, J.P.; Jonaitis, L. Evaluation of the Effectiveness of Helicobacter pylori Eradication Regimens in Lithuania during the Years 2013-2020: Data from the European Registry on Helicobacter pylori Management (Hp-EuReg). Medicina (Kaunas) 2021, 57, doi:10.3390/medicina57070642.
4. Nyssen, O.P.; Bordin, D.; Tepes, B.; Pérez-Aisa, Á.; Vaira, D.; Caldas, M.; Bujanda, L.; Castro-Fernandez, M.; Lerang, F.; Leja, M.; et al. European Registry on Helicobacter pylori management (Hp-EuReg): patterns and trends in first-line empirical eradication prescription and outcomes of 5 years and 21 533 patients. Gut 2021, 70, 40-54, doi:10.1136/gutjnl-2020-321372.
5. Gatta, L.; Nyssen, O.P.; Fiorini, G.; Saracino, I.M.; Pavoni, M.; Romano, M.; Gravina, A.G.; Granata, L.; Pellicano, R.; Gasbarrini, A.; et al. Effectiveness of first and second-line empirical treatment in Italy: Results of the European registry on Helicobacter pylori management. United European Gastroenterol J 2023, 11, 103-113, doi:10.1002/ueg2.12348.
Round 2
Reviewer 2 Report
Comments and Suggestions for Authors
None
Author Response
Reviewer 2 Comment 1:
Please write in italics H. pylori in all the references.
Response: Thank you for your comment. The references have been revised to write H. pylori in italics and changes are highlighted in red in revised manuscript.
Reviewer 2 Comment 2:
It is well known that antimicrobial resistance is a very big problem. So I ask to the authors why on 1000 patients, a laboratory test (culture and susceptibility test, or only molecular test for antimicrobial susceptibility) was performed on 31 cases only? The Maastricht VI/Florence consensus report, well describes this aspect.
Response: We wholeheartedly agree with the reviewer that antimicrobial resistance is a very big problem and that resistance surveillance is important as recommended in the Maastrict Guidelines (reference 8 in manuscript) and our Irish Consensus Guidelines (reference 39 in manuscript) (now mentioned in lines 222-224 of Discussion). Unfortunately, our study on treatment-naïve patients revealed that in the real-world setting of clinical practice in Ireland antimicrobial susceptibly testing is rarely done. It is possible that antimicrobial susceptibility testing is more commonly performed in patients in whom anti-H. pylori treatment was unsuccessful in the Irish Healthcare setting and this will be a focus of our ongoing research in this area.